**Data Availability Statement:** All data are within the manuscript and its Supporting Information files.

**Funding:** C-h Hsieh received funding from the National Center for Theoretical Sciences, Foundation for the Advancement of Outstanding

# Importance of prey size on investigating prey availability of larval fishes

Yu-Hsuan Huang[1], Hsiao-Hang Tao[1]*, Gwo-Ching Gong[2], Chih-hao Hsieh[1,3,4,5]

**1** Institute of Oceanography, National Taiwan University, Taipei, Taiwan, **2** Institute of Marine Environment and Ecology and Center of Excellence for the Oceans, National Taiwan Ocean University, Keelung, Taiwan, **3** Research Center for Environmental Changes, Academia Sinica, Nankang, Taipei, Taiwan, **4** National Center for Theoretical Sciences, Taipei, Taiwan, **5** Institute of Ecology and Evolutionary Biology and Department of Life Science, National Taiwan University, Taipei, Taiwan

* hsiaohang.tao@gmail.com

## Abstract

Prey availability plays an important role in determining larval fish survival. Numerous studies have found close relationships between the density of mesozooplankton and larval fishes; however, emerging studies suggest that small-size zooplankton are more important prey for some larval fish species. One arising question is whether the size of zooplankton determines the relationship between zooplankton and larval fish community in natural environments. To address this question, we collected small-size (50–200 μm) zooplankton, mesozooplankton (> 330 μm), and larval fish using three different mesh-size (50, 330, 1000 μm, respectively) nets in the East China Sea, and examined their relationships in density. Both meso- and small-size zooplankton densities showed positive relationships with larval fish density, while the relationship is much stronger for the small-size zooplankton. Specifically, the smallest size classes (50–75 and 75–100 μm) of small-size zooplankton showed the highest positive relationships with larval fish density. Temperature, salinity, and chlorophyll-a concentration did not significantly explain larval fish density. Based on these findings, we demonstrate the importance of considering prey size when investigating prey availability for larval fishes.

## Introduction

Understanding the key regulating factors of population dynamics is a fundamental question in ecology. In the case of fish populations, fish at larval stage usually suffer extremely high mortality which critically influence the following recruitment variability and further shape the fish population size [1]. The possible factors influencing the mortality of larval fish include predation, environmental perturbations, and starvation (e.g. [2]). In particular, starvation is usually considered as the main cause of larval mortality (e.g. Cushing's "Match-mismatch" hypothesis, [3]). Indeed, some field observations have shown close links between larval recruitment and plankton abundance [4, 5]; however, other studies did not observe such a relationship [6, 7]. One possibility is that the association between larval fish and zooplankton depends on sizes of

Scholarship, and the Ministry of Science and Technology (MOST), Taiwan. H-H Tao received funding from the Ministry of Science and Technology, Taiwan.

**Competing interests:** The authors have declared that no competing interests exist.

zooplankton, yet most of the existing studies did not classify zooplankton community into different size groups when relating them with larval fish community in field studies [4, 6, 8–10].

While many studies have focused on the links between mesozooplankton and larval fish, recent studies have unveiled small-size zooplankton as an important part of the diet for several larval fish species [11–13]. For example, several laboratory studies suggested that protists, as small-size zooplankton, fulfill all or a part of nutrition and energy needs of larvae of Atlantic cod, Pacific herring, and Atlantic herring [14, 15]. A recent empirical study also revealed significant relationships between the abundance of Atlantic herring larvae and microplankton, suggesting strong grazing dynamics among lower trophic levels [11]. In addition, gut content analysis of various marine larval fish species also showed that the most abundant prey taxa are eggs, nauplii, and copepodites, rather than larger zooplankton [16].

Prey selectivity and diet breadth of larval fish depend on the abundance and mouth gape of larval fish, as well as prey availability [13, 16–20]. While these studies examined one or few larval fish species and their suitable prey, it remains unresolved whether zooplankton size determines the general relationships between zooplankton and larval fish community in marine natural environments. Such understanding helps to identify the dominating trophic interactions between marine larval fish and zooplankton. To fill this research gap, we collected different sizes of zooplankton in the East China Sea, and investigated how the zooplankton are associated with the density of larval fish community.

## Materials and methods

### Study area and sampling methods

We carried out our sampling in the East China Sea (ECS) (Fig 1A). This area is an important spawning ground for several fish species, e.g. jack mackerel (*Trachurus japonicus*), Japanese anchovy (*Engraulis japonicus*) [21, 22]. The sampling scheme composed of 8 cruises in total, spanning from 2009 to 2017, and overall 40 cruise-stations. All samples were collected during the warm season (in May and July). We used plankton nets of three different mesh sizes to collect three targeted communities, including 50 μm, 330 μm, and 1000 μm for small-size zooplankton, mesozooplankton, and larval fish, respectively. All nets were towed oblique from the surface to 200 m depth, or 10 m above sea bottom at stations shallower than 200 meters. The nets were equipped with Flow Meter (Hydro-Bios) to record the amount of water passing through. After collection, zooplankton samples were preserved in 5% formalin and larval fish samples were preserved in 95% EtOH in room temperature. We also measured sea surface temperature, salinity, and chorophyll-a concentration at each cruise-station. Temperature and salinity were measured from the conductivity-temperature-depth (CTD) rosette system (Seabird) at each cruise. Measurements at 10m-depth were used to represent the sea surface condition (e.g. [22]). Chlorophyll-a concentration was analyzed from water samples collected by 20L Go-Flo bottles equipped on CTD, following the standard protocols [23]. Mean concentrations of chlorophyll-a were calculated by averaging values from multiple depths above 10 m. No permit is needed to carry out our sampling in these study sites.

### Laboratory work

For each cruise-station, we obtained the density of larval fish, small-size zooplankton, and mesozooplankton (individual m$^{-3}$) by dividing the abundance of each group by filtered water volume. For each small-size zooplankton sample, we counted and identified approximately 150 individuals (with appropriate subsampling using Folsom splitter) for eight taxa, including three taxa (Calanoid, Oithonid, and Harpacticoid) in nauplii stage and five taxa (Calanoid, Oithonid, Harpacticoid, Oncaeid, and Corycaeid) in juvenile stage. We measured the body

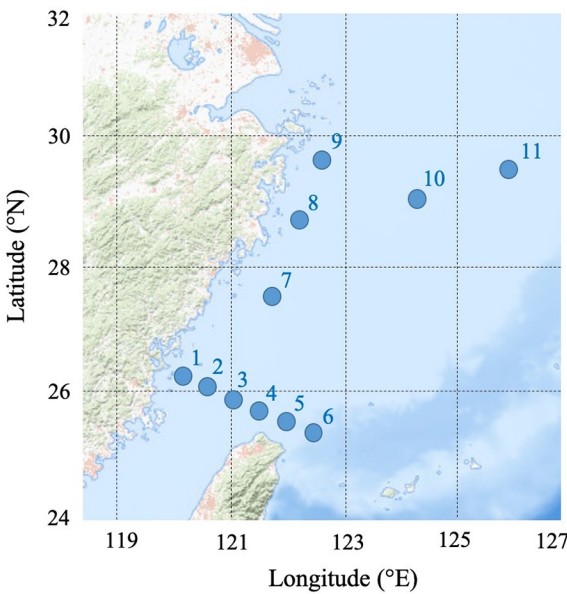
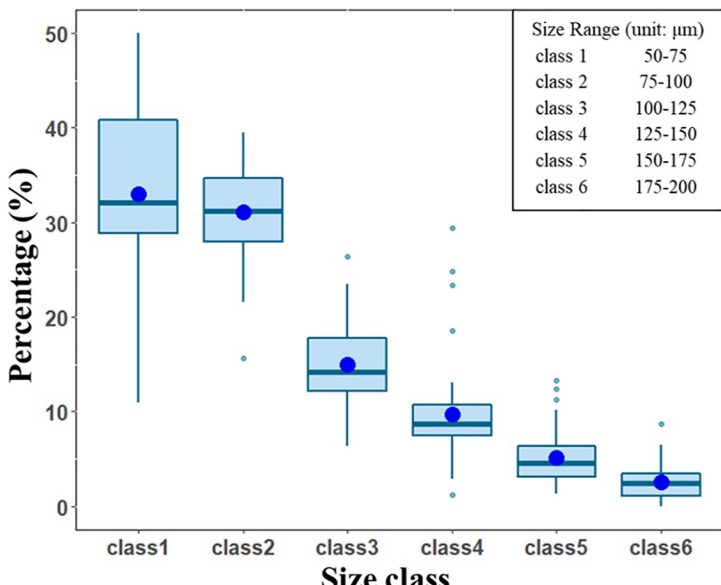

**Fig 1.** (a) Sampling sites in the East China Sea. (b) Size distribution of small-size zooplankton community across cruise-stations. The percentage of each size class was estimated for each cruise-station. Bars within boxes are median, blue dots are average, the upper and lower limits are 75 and 25 quantiles, respectively. Six size class from 1–6 accounted for 33.01%, 31.06%, 14.98%, 9.73%, 5.11%, and 2.58% of total abundance across cruise-stations, respectively. The map used in this figure is obtained from the USGS National Map Viewer (public domain) for illustrative purposes only.

width of each individual under the microscope and omitted individuals bigger than 200 μm. For each mesozooplankton sample, we analyzed approximately 1000 individuals (with appropriate subsampling using Folsom splitter) using the ZooScan integrated system and followed the semi-automatic classification method [24] to identify the taxa. To facilitate comparisons between small-size zooplankton and mesozooplankton, we considered only copepod taxa in mesozooplankton samples (hereafter we used the term "mesozooplankton" and "copepods" interchangeably). The average densities of larval fish, small-size zooplankton, and mesozooplankton were $0.799 \pm 1.39$ ind. m$^{-3}$, $64\,513.63 \pm 124\,525.44$ ind. m$^{-3}$, and $441.95 \pm 495.16$ ind. m$^{-3}$ across cruise-stations. The inter-cruise variation (measured as coefficient of variation) for small-zooplankton, mesozooplankton, and larval fish is 1.43, 0.58, and 0.92, respectively.

We categorized the small-size zooplankton into six size classes according to body width. The six size classes were 50–75 μm, 75–100 μm, 100–125 μm, 125–150 μm, 150–175 μm, and 175–200 μm, respectively (size that falls at the upper limit of each class belongs to the next larger group). The composition of six size classes of small-size zooplankton were on average 33.01%, 31.06%, 14.98%, 9.73%, 5.11%, and 2.58% across cruise-stations (Fig 1B). Raw data of larval fish and zooplankton densities are available in S1 and S2 Tables. Our study does not need a review by an animal ethics committee.

## Data analyses

To analyze the relationship of larval fish density versus zooplankton density and environmental variables, we fitted our data with linear mixed-effect models (LMM). The zooplankton density of each size class or environmental factor was included in each separate model as the explanatory variable (fixed effect), while the response variable in each model was larval fish density. Cruise was considered as a random effect, to avoid spurious correlation due to among-cruise variation. Before the fitting, we log-transformed all the density data for normality. We used "lme4" package for LMM analysis in the R software.

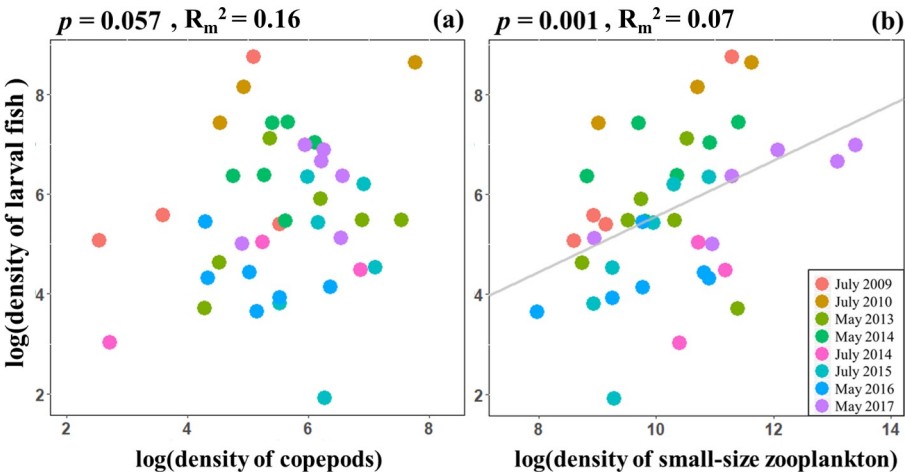

**Fig 2. Relationship between prey and larval fish density (ind. m$^{-3}$) at log-log scale.** Density of copepods exhibited a positive but not statistically significant relationship with larval fish density (a), whereas small-size zooplankton showed a significant relationship with larval fish density (b). Colors indicate different cruises; grey line in (b) indicates a significant regression from the linear mixed-effects model. *P*-values and marginal R-squared values are reported.

## Results

Both small-size zooplankton and copepod densities showed positive relationships with larval fish density under log-log scale ($p = 0.001$ and $p = 0.057$, respectively; Fig 2, S3 Table). Moreover, the density of small-size zooplankton exhibited a much stronger relationship with larval fish density than that of copepods. In addition, the positive relationship between larval fish density and the density of small-size zooplankton were observed in most cruises, whereas the relationships with copepods were inconsistent among cruises; some cruises showed hump-shaped (i.e. May 2013) or even negative relationships (i.e. July 2016).

When dividing small-size zooplankton into six size classes, densities of 50–75 μm, 75–100 μm, 100–125 μm, and 150–175 μm (size class 1, 2, 3 and 5) showed positive relationships with the larval fish density (Fig 3A–3C and 3E, S3 Table). Specifically, the relationships between the two smallest size classes (50–75 μm and 75–100 μm) of zooplankton and the larval fish density were highly significant ($p < 0.001$ and $p = 0.001$; Fig 3A and 3B, S3 Table). Sizes above 100 μm appeared to show positive relationships with larval fish density in most of cruises, yet the relationships were relatively weak ($p = 0.051$, $p = 0.048$, and $p = 0.368$ for 100–125 μm, 125–150 μm, and 150–175 μm of size class 3, 4, and 5; Fig 3C–3E, S3 Table). The largest small-size zooplankton (175–200 μm) had the lowest density among all size classes and did not explain the larval fish density (Fig 3F, S3 Table). Temperature, salinity, and chlorophyll-a concentration did not significantly explain larval fish density over the sampling period (Fig 4).

## Discussion

The density of small-size zooplankton best explained larval fish density at our study sites, compared to mesozooplankton (Fig 2). Specifically, the smallest classes of small-size zooplankton (50–75 μm and 75–100 μm for size class 1 and 2, respectively) exhibited the strongest positive relationships with larval fish density, compared to other size classes (Fig 3). These results suggest that small-size zooplankton are likely a more important prey for the larval fish community in the East China Sea during the sampling period.

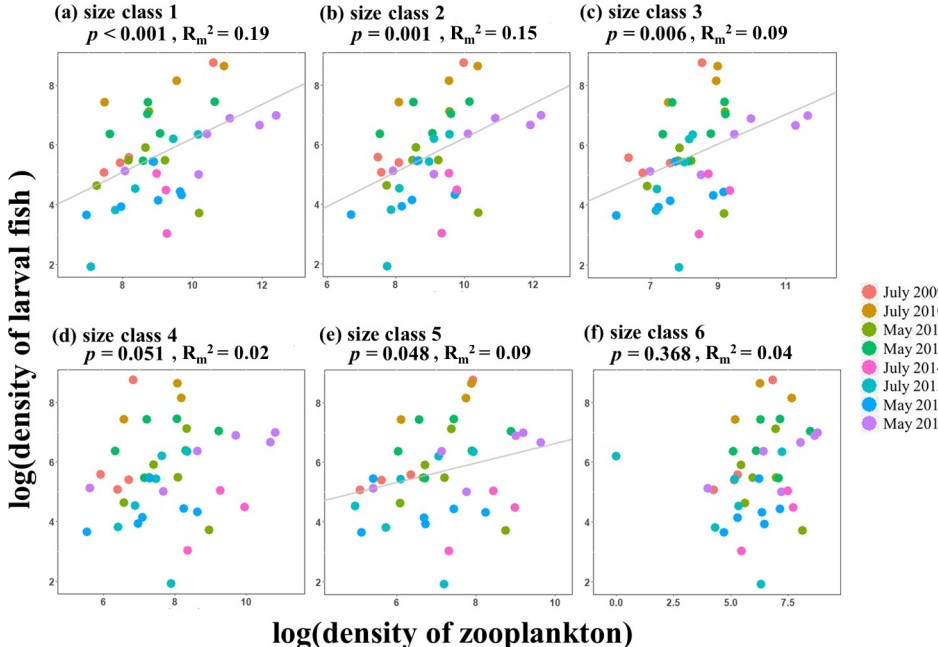

**Fig 3. Relationships between densities of six size classes of small-size zooplankton and larval fish density at log-log scale.** Colors indicate different cruises; grey lines indicate significant regression from linear mixed-effects models. *P*-values and marginal R-squared values are reported.

The positive relationships between larval fish and two sizes of zooplankton community suggest bottom-up ecosystem trophic dynamics in the Eastern China Sea. This finding corroborates with some other marine ecosystems where bottom-up processes dominate; e.g. Northeast Pacific [25]. Our results also support a previous finding in the northern Taiwan Strait, where the spatial distribution of larval community is positively linked with copepod density [4]. Another study in the southwest Nova Scotia also showed positive relationships between growth conditions of larval gadoid and zooplankton abundance [8]. Our results differ from empirical observations in other regions, where fish recruitment negatively correlates with zooplankton abundance (e.g. [6, 7]).

Importantly, we showed that the density of larval fish community is better explained by small-size zooplankton (50–200 μm), rather than by mesozooplankton (> 330 μm). Most of

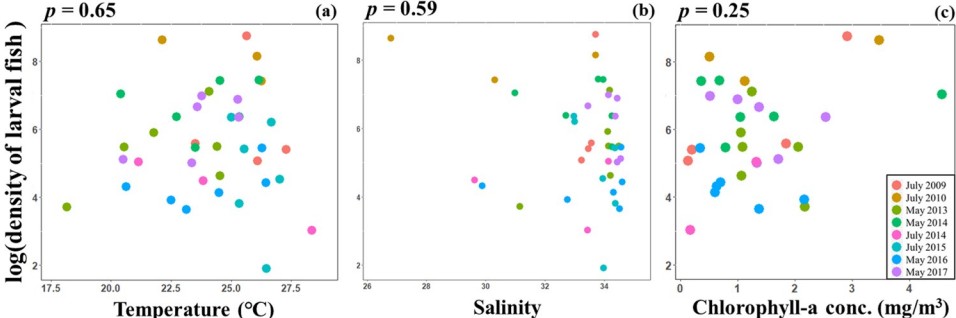

**Fig 4. Relationships between environmental variables and logged larval fish density.** The environmental variables included (a) temperature, (b) salinity, and (c) chlorophyll-a concentration at the surface layer. Colors indicate different cruises. None of the relationships were significant.

the existing studies found small-prey size preferences for specific larval fish species. For example, at the coastal waters of northern Norway, small copepodites (i.e. *Acartia spp.* and *Temora longicornis*) spatially co-occurred with Capelin larvae, while bigger prey organisms (i.e. Copepod nauplii and *Calanus finmarchicus*) were less abundant at these locations [9]. In another study in Ohio reservoirs, the mean prey size of gizzard shad (*Dorosoma cepedianum*) larvae did not increase with larvae size, suggesting that small-size zooplankton as its main prey source [10]. To the best of our understanding, our work is the first to show that the relationship between zooplankton and larval fish community depends on the size of zooplankton in the marine natural environment.

One may argue that the positive relationship between small-size zooplankton and larval fish density in our study may arise due to their co-occurrence driven by water currents. In other words, water currents may bring most of planktonic organisms and larval fish to the same locations. If so, both small-size and mesozooplankton would have shown a similar strength of positive relationship with larval fish density. However, in our study, only the density of small-size zooplankton exhibited a significant positive relationship with larval fish density (Fig 2). This evidence suggests that our findings are not due to co-occurrence of these organisms.

In our study, larval fish density does not appear to associate with any of the environmental variables (Fig 4). The lack of relationship between sea surface temperature and larval density may be due to the limited sampling seasons. Specifically, our study area is located in a subtropical region, and all of the sampling were carried out in warm seasons (May and July). In contrast to temperate regions where temperature is often a limiting factor for larval growth (e.g. [17]), the temperature range in our study area is likely to be suitable for larval growth and therefore less influential on larval fish density. The lack of relationship between chlorophyll-a concentration and larval fish density suggests that chlorophyll-a concentration does not indicate food availability of larval fish in our study area. Interestingly, chlorophyll-a concentration has a positive relationship with mesozooplankton density, but exhibits a none-significant relationship with small-size zooplankton is lacking (S1 Fig). It is likely because small-size zooplankton consume not only phytoplankton but also protists. Thus, the sources of food could come through not only grazing food chain but also microbial loop, which could not be fully captured by chlorophyll-a concentrations. Importantly, mesozooplankton density does not have a significant relationship with larval fish density (Fig 2), explaining the lack of relationship between chlorophyll-a concentration and larval fish density. Furthermore, Chen *et al.* (2014) found that the different dominant larval taxa in the East China Sea (e.g. jack mackerel, Japanese anchovy) exhibited different responses to the sea surface salinity. This may explain why there is no clear association between larval fish density and sea surface salinity.

The close link between the larval fish community and small-size zooplankton found in our study is possibly due to two mechanisms: first, the dominant species of the larval fish community prefers small-size prey. Second, the majority of sampled larval fish have small body size (or mouth gape size), which limits the intake of bigger prey [20, 26]. To test these hypothesized mechanisms, future work could focus on examining size and taxa of larval fish, and their gut content. Using these data, we can further estimate the size ratio between larval fish and its prey, in order to identify the optimal prey size (e.g. [27]). Moreover, the knowledge on high-resolution taxonomical information of prey can help examine prey selectivity of larval fish [28].

In summary, our results from the East China Sea showed that the density of small-size zooplankton (nauplii and copepodites), especially at the size range of 50–100 μm, best explained the larval fish density. These findings suggest the importance of considering prey size when investigating prey availability of larval fishes.

## Supporting information

**S1 Table. Densities of zooplankton, larval fish, and environmental variables of each cruise-station.**
(DOCX)

**S2 Table. Size composition of small-size zooplankton of each cruise-station.**
(DOCX)

**S3 Table. The results of linear mixed effect models linking larval density with zooplankton density and environmental factors.**
(DOCX)

**S1 Fig. Relationships of chlorophyll-a concentration versus log-transformed density of small-size zooplankton, mesozooplankton, and larval fish.**
(DOCX)

## Acknowledgments

We thank Yo-Ching Lee, Yu-Chu Lin, and crews of R/V Ocean Research I, II, and V for sampling.

## Author Contributions

**Conceptualization:** Yu-Hsuan Huang, Hsiao-Hang Tao.

**Data curation:** Yu-Hsuan Huang, Gwo-Ching Gong.

**Formal analysis:** Yu-Hsuan Huang.

**Funding acquisition:** Chih-hao Hsieh.

**Investigation:** Yu-Hsuan Huang, Gwo-Ching Gong.

**Methodology:** Yu-Hsuan Huang, Chih-hao Hsieh.

**Project administration:** Chih-hao Hsieh.

**Supervision:** Chih-hao Hsieh.

**Validation:** Hsiao-Hang Tao, Chih-hao Hsieh.

**Writing – original draft:** Yu-Hsuan Huang.

**Writing – review & editing:** Hsiao-Hang Tao, Chih-hao Hsieh.

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
