## [Decision Letter · Decision Letter 0]

12 Jan 2021

PONE-D-20-34920

Importance of prey size on investigating prey availability of larval fishes

PLOS ONE

Dear Dr. Hsiao-Hang Tao

Thank you for submitting your manuscript to PLOS ONE. After careful consideration, we feel that it has merit but does not fully meet PLOS ONE’s publication criteria as it currently stands. Therefore, we invite you to submit a revised version of the manuscript that addresses the points raised during the review process.

We have got your manuscript reviewed by four different reviewers. Except one, all three reviewers feel that your manuscript has requires major revision . I personally feel that this Ms. is  a well-executed work collecting comprehensive data on fish larval abundance and zooplankton and mesoplankton over a period of 8 years. Such field studies provide corroborative evidence for the importance of prey size-larval survival and growth relationship established only in laboratory studies. However the manuscript need major revision .

The paper needs thorough investigation for publication. The larval feeding in fish is stage dependent; most fishes do not start external food unless they are ready for it. They do survive with yolk material available on them. The mouth gap of larval fish and prey size are most crucial that determines the relationship between the size fractionation of zooplankton for larval feeding. The present study dealing with relationship between the abundance of fish larvae and size fractionation of zooplankton does not yield enough for publication. The composition of fish larvae as well as their abundance have not been analyzed. This is important as the mouth gap will vary depending upon the species, and therefore a definite relationship could be established. 

Please take in account of comments given by  reviewers and revise the manuscript accordingly.  

We look forward to receiving your revised manuscript.

Kind regards,

Ram Kumar, Ph.D.

Academic Editor

PLOS ONE

Journal Requirements:

3.We note that Figure(s) 1 in your submission contain map images which may be copyrighted. All PLOS content is published under the Creative Commons Attribution License (CC BY 4.0), which means that the manuscript, images, and Supporting Information files will be freely available online, and any third party is permitted to access, download, copy, distribute, and use these materials in any way, even commercially, with proper attribution. For these reasons, we cannot publish previously copyrighted maps or satellite images created using proprietary data, such as Google software (Google Maps, Street View, and Earth). For more information, see our copyright guidelines: http://journals.plos.org/plosone/s/licenses-and-copyright.

a)    You may seek permission from the original copyright holder of Figure(s) 1 to publish the content specifically under the CC BY 4.0 license. 

Reviewers' comments:

Reviewer's Responses to Questions

**Comments to the Author**

1. Is the manuscript technically sound, and do the data support the conclusions?

Reviewer #1: Yes

Reviewer #2: Yes

Reviewer #3: Yes

2. Has the statistical analysis been performed appropriately and rigorously? 

Reviewer #1: Yes

Reviewer #2: Yes

Reviewer #3: Yes

3. Have the authors made all data underlying the findings in their manuscript fully available?

Reviewer #1: Yes

Reviewer #2: Yes

Reviewer #3: Yes

4. Is the manuscript presented in an intelligible fashion and written in standard English?

Reviewer #1: Yes

Reviewer #2: Yes

Reviewer #3: Yes

5. Review Comments to the Author

Reviewer #1: This is an interesting area of research

1. The title should be modified.

2. Abstract - line 22, Modify this sentence.

3. Introduction and Discussion - Recent literature should be consulted. One reference 2017, others are before this period.

4. Materials and Methods - line 62 - The English should be modified. This is also applicable for other places.

5. Line 166, it should not be italic.

The language requires improvement.

This paper may be accepted after minor revision.

Reviewer #2: This essentially a field study is a well-executed work collecting comprehensive data on fish larval abundance and zooplankton and mesoplankton over a period of 8 years. Such field studies provide corroborative evidence for the importance of prey size-larval survival and growth relationship established only in laboratory studies.

Authors may please take into account the points I elaborated below while preparing a revision.

A. Minor errors that need to be corrected:

1. Line 81: Word given here is 'medium' Shouldn't it be 'median' not medium?

2. Lines 101-103. Size intervals for small zooplankton. The same number cannot be in two different size

intervals. (50-75, 75-100). Correct way- Please give size intervals as 50-74, 75-99, 100-124, 125-149,

150-174, 175-200.

3. Line 124. Not correct to say 'marginally' significant (statistically, some thing is either significant or it is not).

I suggest replacing the sentence with " they were positive but not statistically significant (p~0.057)"

4. Line 145. The statement " All relationships were not significant" might imply that some relationships were

significant. To be unambiguous, please state that "None of the relationships were significant".

B. I strongly recommend the following inclusions:

1. Fig. 2 and 3- For each graph the p value is given. But 'r-square (Coefficient of determination) is a more

readily interpretable parameter. I recommend giving r-square values within each graph of Figs. 2 and 3.

2. Size frequencies of larval fishes in the samples. Larval fishes were collected using a net of pore size

1000mum. That means, I suppose that the larvae were all larger than 1000um. However, there is no

information given about the size frequencies of the fish larvae in collected samples. I recommend that

the authors include a figure (as in Fig.1b) showing the size frequencies of larval fish (as they did for small

zooplankton in Fig.1b).

C. Points or questions that need to be clarified:

1. As stated by the authors (and shown as coloured circles in Figs. 2 and 3), data from all the cruises during

the sampling period were combined for examining larval abundance-prey food relationships. However,

considering that the total sampling period spans ~8 years, the reader might wish to have some idea about

the "inter-cruise" variation in fish larval abundance and, if possible, in small and mesozooplankton

abundance also in the sampled areas. At the appropriate place in the text, authors could provide

intercruise variation as a 'coefficient of variation' (C.V.).

2. Relationship between mesozooplankton (copepods?) and larval abundance. It was positive but not

statistically significant. But p value is 0.057, so close to significance level of 0.05). Does this suggest that

the largest size class of the fish larvae might be feeding preferentially on mesozoolankton? Also, I note

that the largest small zooplankton size classes (sizes 5 and 6, Fig.1b) formed less than 10% of zooplankton

abundance.

3. Relationship with chlorophyll. I assume that the dominant food of small zooplankton included green algae

and diatoms, whose densities are generally reflected in chlorophyll concentration in the water. Therefore, I

would have expectd indirectly positive correlations among cholorophyll- zooplankton, and larval fish

abindance. Authors may add a few lines giving probable explanation.

D. Information or data on prey sizes consumed by larval fish

The authros rightly pointed out in the Discussion part (Lines 195-200) the importance of knowing the sizes of zooplankton consumed by the larvae for understanding the mechanism underlying the oserved relationship between larval abundance and zooplankton densities. I wish to know... is it possible that the authors could conduct a gross gut content analysis of a sample of larval fish that they had collected and preserved? Even gape size measurements of preserved larvae may not be too difficult or time-consuming. These two pieces of data, if collected and included in the paper, will strengthen the arguments that they cited in Discussion. I strongly urge the authors to explore the feasibility of accomplishing the additional work thatI suggested.

Reviewer #3: The research paper is coherently created. However inadvertent grammatical errors have to be rectified. For example:

In addition, the positive relationship between larval fish density and the density of

120 small-size zooplankton were observed in most cruises, whereas the relationships with copepods were

121 inconsistent among cruises; some cruises showed hump-shaped (i.e. May 2013) or even negative

122 relationships (i.e. July 2016).

"The positive relationship" has to be followed with "Was" and not "were"

6. PLOS authors have the option to publish the peer review history of their article (what does this mean?). If published, this will include your full peer review and any attached files.

Reviewer #1: **Yes: **Prof. JaiGopal Sharma

Reviewer #2: No

Reviewer #3: No

---

## [Author Response · Author response to Decision Letter 0]

9 Feb 2021

Please see our point-by-point responses to the Editor and reviewers' comments in the Response Letter, included in the revised documents.

---

## [Decision Letter · Decision Letter 1]

26 Apr 2021

Importance of prey size on investigating prey availability of larval fishes

PONE-D-20-34920R1

Dear Dr. Hsiao-Hang Tao 

We’re pleased to inform you that your manuscript has been judged scientifically suitable for publication and will be formally accepted for publication once it meets all outstanding technical requirements. We thank you for the  revision and considering all comments very appropriately. Overall, we are quite satisfied with the submitted revision that clearly shows that all my queries have been answered and or considered. 

Kind regards,

Ram Kumar, Ph.D.

Academic Editor

PLOS ONE
---

## [Editor Report · Acceptance letter]

7 May 2021

PONE-D-20-34920R1 

Importance of prey size on investigating prey availability of larval fishes 

Dear Dr. Tao:

I'm pleased to inform you that your manuscript has been deemed suitable for publication in PLOS ONE. Congratulations! Your manuscript is now with our production department. 

Kind regards, 

on behalf of

Professor Ram Kumar 

Academic Editor

PLOS ONE